# Nanostructured Glass-Ceramic Materials from Glass Waste with Antimicrobial Activity

**DOI:** 10.3390/molecules29133212

**Published:** 2024-07-06

**Authors:** Juliani P. Caland, João Baptista, Gabrielle Caroline Peiter, Kelen M. F. Rossi de Aguiar, Horácio Coelho-Júnior, João P. Sinnecker, Jorlandio F. Felix, Ricardo Schneider

**Affiliations:** 1Núcleo de Física Aplicada, Instituto de Física, Brasília, Universidade de Brasília-UnB, Brasilia 70910-900, DF, Brazil; julianicaland96@hotmail.com; 2Group of Polymers and Nanostructures, Universidade Tecnológica Federal do Paraná—UTFPR, Toledo 85902-490, PR, Brazil; 3Brazilian Center for Physics Research, Rio de Janeiro 22290-180, RJ, Brazil; horaciocoelhojunior@gmail.com (H.C.-J.);

**Keywords:** glass waste, glass-ceramic materials, antimicrobial materials

## Abstract

Modern consumption patterns have led to a surge in waste glass accumulating in municipal landfills, contributing to environmental pollution, especially in countries that do not have well-established recycling standards. While glass itself is 100% recyclable, the logistics and handling involved present significant challenges. Flint and amber-colored glass, often found in high quantities in municipal waste, can serve as valuable sources of raw materials. We propose an affordable route that requires just a thermal treatment of glass waste to obtain glass-based antimicrobial materials. The thermal treatment induces crystallized nanoregions, which are the primary factor responsible for the bactericidal effect of waste glass. As a result, coarse particles of flint waste glass that undergo thermal treatment at 720 °C show superior antimicrobial activity than amber waste glass. Glass-ceramic materials from flint waste glass, obtained by thermal treatment at 720 °C during 2 h, show antimicrobial activity against *Escherichia coli* after just 30 min of contact time. Laser-induced breakdown spectroscopy (LIBS) was employed to monitor the elemental composition of the glass waste. The obtained glass-ceramic material was structurally characterized by transmission electron microscopy, enabling the confirmation of the presence of nanocrystals embedded within the glass matrix.

## 1. Introduction

Glass materials, such as window panes, bottles, household goods, and lamps, are widespread in modern life. Many of these applications of glass come from the characteristics noticed in ordinary glass based on silica, such as transparency, chemical resistance, and its affordability. Among commercial glass, 95% of the glass produced is estimated to contain oxides, with a major concentration of SiO_2_, the so-called silicates. On the other hand, the wide use of silicate glass in the most diverse applications has led to an environmental problem associated with their post-consumer use [1,2]. Although silica-based materials have ideal properties for recycling, their disposal for landfilling is inappropriate due to their high chemical stability producing a non-biodegradable material [3]. Beyond the environmental problem of glass materials’ disposal, sand, the raw mineral in glass synthesis, is being mined almost uncontrollably. Further, less than 1% of the world’s land is suitable for industry, and the extraction of sand impacts the environment [4].

The chemical stability, widespread use, and well-established synthesis method of glass present a challenge in municipal waste management, primarily due to the low added value associated with waste glass. The logistics for glass re-fusion facilities are not economically viable due to transport distances, legislation, and collection costs [5]. Thus, it is necessary to find local or regional alternatives to glass waste, mainly in countries with continental dimensions [3].

Waste glass can be found in municipal landfills with different compositions since glass compositions are designed to exhibit different optical, physical, and chemical properties. Usually, colored bottles are used to prevent product oxidation, e.g., beer and wine, and extend the product shelf life during storage. However, the bottle/container can be colored due to aesthetic reasons too. Amber-colored and flint glass are the most common types of glass found in domestic waste and municipal landfills.

Every day, huge amounts of waste glass can be found in municipal landfills. The classical method for glass recycling is to melt the glass to obtain new glass-based materials with virtually no losses during the process. However, the logistics (e.g., transport, freight, and toll) for industries can be a drawback for recycling. Not only is the development of strategies for the reuse of waste glass as new materials desirable [1,6,7], but also, adding functional properties to waste-based material is important. Punj et al. [8] highlighted the application of agro-food wastes for bioceramics and bioglass synthesis, which could be an application where consumption would be significant. They showed that ceramic biomaterials derived from agro-food waste/ashes have various advantages over synthetic ceramic biomaterials, such as minimal tissue rejections, good biocompatibility, and biodegradability. In this sense, the antimicrobial character of glass is unusual. The synthesis of antimicrobial glass/glass-ceramic from ordinary silica-based glass can show the chemical resistance of the pristine material. Moreover, glass shows high thermal and chemical resistance with low cost and can be obtained at a large scale worldwide. The relatively low cost of glass materials, principally after their use, results in a low income generated by recycling. Thus, the availability of the glass in waste becomes an opportunity for the development of new materials/functionalities in open-loop recycling.

The biocide activity of the silica-based bioactive glass-ceramic substrate, 45S5 Bioglass^TM^, was reported by Cabal et al. [9]. The authors noticed that glass-ceramic scaffolds could act directly against common bacteria without metallic nanomaterials such as silver or copper coatings [9,10]. In this case, only the combeite phase (Na_2_Ca_2_Si_3_O_9_) showed antimicrobial activity due to the calcium release in the medium. However, the authors also argued that the calcium-rich glass phase can be involved in antimicrobial activity [10]. Moya et al. reported the synthesis of calcium oxide-rich glass with antimicrobial activity from environmentally friendly raw materials [11]. Soda-lime glass-ceramics were successfully used to obtain biocompatible materials for dental applications that can prevent implant infections [12]. Furthermore, in dentistry, root canal filling is performed with tricalcium silicate-based materials that not only promote antimicrobial activity but also act as a bioactive material that can induce the formation of hard tissue. As a general trend, calcium release is positively correlated with enhanced antimicrobial performance [13,14]. Usually, antimicrobial activity is achieved by the modification or synthesis of designed glass compositions, which can be a drawback for some applications [10,15]. The development of a solvent-free route from inexpensive raw materials or waste without the addition of transition metals (e.g., silver or copper) is highly desirable [16].

In this work, we have proposed an affordable route to convert powdered waste glass into glass-ceramic materials with antimicrobial activity. Two types of waste glass, namely amber-colored and flint, were submitted to a crystallization process through thermal treatment at 720 °C, resulting in the formation of two distinct nanocrystalline phases: nepheline and combeite. The nanosized crystalline phases obtained on the powder glass surface were evaluated for their antimicrobial performance against *Escherichia coli*. The glass-ceramic material originating from flint glass waste showed antimicrobial activity after only 30 min of contact time. Therefore, the proposed approach overcomes several drawbacks to synthesizing antimicrobial materials using only a thermal treatment and ordinary recycled glass as raw materials. The glass-ceramics were characterized by High-Resolution Transmission Electron Microscopy (HRTEM), Selected Area Electron Diffraction (SAED) allowing dark-field (DF) TEM imaging, and Energy-Dispersive X-ray Spectroscopy (EDS) (EDS) mapping in STEM (Scanning Transmission Electron Microscopy) mode. Additionally, the elementary composition and particle size distribution of powdered waste glass was evaluated by Laser-Induced Breakdown Spectroscopy (LIBS) and static laser scattering (SLS), respectively.

## 2. Results and Discussion

The elemental composition of the flint glass, i.e., Si, Ca, Mg, Na, and Al, shows remarkable similarity with amber glass. Figure 1 shows the LIBS analyses of the sintered amber waste glass (Figure 1 line (I)) and flint waste glass (Figure 1 line (II)). The difference between the two studied glass is the presence of Ba (553.5 nm, 493.4 nm, and 455.4 nm lines) in amber glass. The elemental composition was also determined by X-ray fluorescence, which showed that flint glass is highly similar to amber glass, as shown in Appendix A.

The presence of primary elements, i.e., Si, Ca, Na, and Al, in amber and flint waste glass enables us to theorize that these types of glass can crystallize similarly (Figure 1), thus forming a glass-ceramic with antimicrobial properties, as will be shown later. Coarse and fine particles for amber-colored and flint glass (Appendix A) were evaluated against the antimicrobial properties after thermal treatment.

With the aim of obtaining an understanding of the antimicrobial activity, structural and chemical electron microscopy analyses were performed in the glass-ceramic materials from waste. Powder X-ray diffraction analyses of both annealed (Appendix A) and unannealed (Appendix A) glass powder waste did not reveal any peaks indicative of glass crystallization, probably due to the low abundance of crystalline phases. Figure 2a displays an ADF-STEM micrograph of the flint glass sample annealed at 720 °C for 2 h. It is possible to observe the presence of nanostructures morphologically characterized by brighter contrast regions embedded in a glass matrix. A set of EDS mapping shows the elements associated with the combeite phase [Ca_4_Na_4_O_18_Si_6_, (ICSD collection code: 031246)], as well as its distribution in the region analyzed. The SAED pattern obtained from this region (Figure 2b) shows the amorphous character of the sample with a substantial presence of diffraction spots (highlighted in white arrows). After all this, Figure 2c shows the nanoparticles in detail, making it possible to identify interplanar spacing of 0.20 nm, which matches the d-spacing of (043) crystallographic planes attributed to the combeite phase. However, the (104) family planes from the nepheline phase [Al_6.24_Na_6.65_O_32_Si_9.76_, (ICSD collection code: 200584)] have interplanar spacing of 0.20 nm, which does not exclude the possibility of reinterpretation.

Although there was antimicrobial activity of the glass-ceramic after thermal treatment for two hours, the flint waste glass was submitted to five hours of thermal treatment at 720 °C to increase the portion of the crystallized phases, thus improving the signal of the diffracted beam to evince the obtained phases. Figure 3a shows TEM measurements where we can see the crystalline morphology, and in Figure 3b, it is possible to see the crystalline character of the nanostructures. Here, we have an SAED pattern, different from Figure 2b, clearly showing that the thermal treatment during 5 h was more efficient for the nucleation of the nanostructures. In Figure 3b, we also drew semirings (triple lines in yellow) based on theoretical lattice parameters. The (024), (13$\bar{2}$), and (24$\bar{2}$) family planes correspond at the combeite phase, corroborating the theoretical values of 0.268, 0.234, and 0.165 nm, respectively. In addition, the family planes (153), (434), and (605) are indexed exclusively to the nepheline phase, with the following respective values measured: 0.135, 0.117, and 0.109 nm. We suggest the combeite and nepheline phases are grown after thermal annealing, as noticed by Cabal et al. [10]. In fact, in Figure 3c,d, we show the morphology from structures formed with (24$\bar{2}$) and (605) family planes, where the micrographs were obtained by choosing the respective spots from Figure 3b, indicated by the solid red circles. Figure 3c,d are dark-field TEM micrographs (DF 24$\bar{2}$ and DF 605, respectively, from combeite and nepheline phases). In particular, in Figure 3d, the dark-field image (DF 605) shows the nepheline phase structure embedded in the sample. Also, we have well-defined spots (and not ring patterns) in the region correspondent to (605) family planes in Figure 3b, showing that the presence of nanostructures with the nepheline phase may be outnumbered or morphologically in larger sizes, shown in a comparison between Figure 3c,d.

Finally, Figure 4a is an HRTEM image taken from the image upside area shown in Figure 3a, exhibiting the interplanar spacing from synthesized nanostructures. Figure 4b is a higher-magnification image from a Figure 4a region (indicated by the continuous triple line in violet color). It shows the 0.23 nm and 0.16 nm lattice spacing attributed to the combeite phase, while 0.10 nm is the interplanar spacing of the nepheline phase. This corroborates with the SAED analysis (Figure 3b) that shows combeite and nepheline are coexisting. Moreover, the size of the nanoparticles increases with the time of thermal treatment (please refer to Figure 2c and Figure 4a, showing that the sample was treated with 2 h of thermal treatment and started the nucleation process, while thermal treatment for 5 h was more favorable to increase the crystallization of the sample and to promote phases’ coexistence).

Based on the results of the antimicrobial activity assay (Table 1), samples AGT1, AGT2, PGT1, AFT2, and PFT2 show antimicrobial activity (3-log reduction after 24 h) against *E. coli*. At the same time, a complete reduction in colonies was noticed with the PGT2 sample. Thus, the broth dilution experiments (Figure 5) were performed only for the PGT2 sample with a contact time of 30 min. Figure 5a shows the results of bacterial growth of *E. coli* ($10^{6}$ CFU) in a liquid medium after 30 min. The bacterial growth can be evaluated by the optical density change as a function of analysis time (Figure 5b). The proposed process allows the production of glass-ceramic material with inhibitory capacity for bacteria growth within 30 min of contact (Figure 6).

According to some authors [10,17,18,19,20], antimicrobial activity is associated with the local change in pH due to the dissolution of bioactive glass. Cabal et al. [9] showed that, following antimicrobial tests, there was an increase in the pH of the medium. However, it was found that this increase was not significantly greater than the pH recorded in the fermentation broth at the end of the growth period. Therefore, based on their experiments, it was impossible to establish a conclusive relationship between the biocidal activity and the pH of the medium. On the other hand, Moya et al. [11] attributed the biocidal activity to the presence of high concentrations of calcium ions at the glass-ceramic–bacteria interface. During the active growth stage, the cell membrane is positively charged. The interaction between the negatively charged glass particles and the cell walls occurs at this stage. In the second step, leaching of Ca^2+^ ions from the surface of the glass particle occurs only at the glass–bacterial membrane interface, causing cell death (Figure 7). This effect was evidenced when Cabal et al. evaluated coated and uncoated Bioglass™ 45S5, and both released the same amount of calcium ions, reaching identical biocidal activity [9,21].

The antimicrobial activity of bioactive glass has been extensively studied due to the use of silicate-based glass for bone regeneration. The classical approach of adding silver to the glass compositions as bioglass can provide antimicrobial activity from Ag ions released during the dissolution of bioactive glass [18,22], or soda-lime [23] compositions. Adding non-noble metals such as copper [24] and zinc [25] provides an alternative to producing glass materials with antimicrobial activity. For doped glass, ion release is required, although the calcium ions improve and/or show a synergistic effect on the antimicrobial activity [23,24]. The glass matrices without doping with silver or copper, which show some antimicrobial activity, have a performance related to the calcium ions [10,11]. For the waste glass sintered in this work, calcium is a common addition to improve the chemical resistance of soda-lime glass. Thus, as will be demonstrated in further discussion, antimicrobial activity is associated with the partial crystallization of the glass, in which the thermal treatment induces the formation of the combeite phase [26]. Indeed, Cabal et al. [10] showed that the presence of the combeite phase together with the glass matrix facilitates the release of calcium ions (Figure 7), which in turn leads to membrane depolarization and the subsequent death of the cells [10].

## 3. Methods

### 3.1. Antimicrobial Glass-Ceramic Synthesis

Waste glass from a municipal landfill (Toledo City, Paraná State, Brazil) was collected and cleaned exhaustively with water. The bottles were manually crushed in an agate mortar and sieved in sieves of mesh (48–100) and (100–200). After that, the powder was cleaned with HCl (2 M) for 45 min and washed with distilled water until pH 6–7 was achieved. The powder was dried under an air atmosphere at 120 °C for two hours. Disks of approximately 150 mg of glass were obtained by sintering the powdered glass in a steel mold (8 mm diameter and 3 mm height). The thermal treatment was selected by performing a screening of the temperatures usually employed for the crystallization of silica-based glass [10,27]. Screening was performed, varying the glass type (amber or flint) and granulometry ((48–100) or (100–200)), with the thermal treatment of 700 °C or 720 °C at a fixed time of 2 h. After thermal treatment, the sintered glass-ceramic discs were crushed and used for the antimicrobial assays.

### 3.2. Morphological and Spectroscopy Characterizations

High-Resolution Transmission Electron Microscopy (HRTEM), Selected Area Electron Diffraction (SAED) allowing dark-field (DF) TEM imaging, and Energy-Dispersive X-ray Spectroscopy (EDS) were performed through a JEOL FEG JEM 2100F transmission electron microscope (TEM) (JEOL, Tokyo, Japan) operated at 200 kV of acceleration voltage and equipped with an EDS detector (Bruker XFlash 6T-60, Bruker, Billerica, MA, USA). The Scanning Transmission Electron Microscopy (STEM) mode image was obtained using an annular dark-field (ADF) detector. The powder X-ray diffraction (PXRD) was performed using the Rigaku diffractometer model SmartLab SE with Cu-K_α_ radiation (λ=1.5418 Å) between 10° and 70° (θ−2θ) (Bragg–Brentano geometry), collecting the signal using a 1D D/tex Ultra 250 detector (Rigaku, Tokyo, Japan).

The samples were prepared as follows: Powdered glass samples were immersed in isopropyl alcohol followed by sonication for 30 min and dispersed in Ni holey carbon grids (Quantifoil Q25035 R2/1 200M, Quantifoil, Großlöbichau, Germany).The dispersed samples were dried at room temperature for 24 h and used for Transmission Electron Microscopy measurements.

The laser-induced breakdown spectroscopy (LIBS) measurements were performed using a J200 Tandem LIBS spectrometer from Applied Spectra (West Sacramento, CA, USA) operating with a 266 nm laser (25 mJ nanosecond and pulse width (FWHM) < 6 nsec) and equipped with a 6-channel charge-coupled device (CCD) spectrometer with spectral coverage from 190 nm to 850 nm and resolution better than <0.1 nm. The measurements were performed using a grid with 30 points (5 × 6), laser power operating at 25%, gate delay of 1.0 μs, 10 shots, and a spot size of 35 μm. The analyses were performed with three different disks. The spectra obtained were analyzed using the National Institute of Standards and Technology (NIST) database [28]. All experiments were performed under an air atmosphere. The X-ray fluorescence analysis was performed using an AXIOS mAX-Advanced from Panalytical (Malvern Panalytical, Malvern, UK).

### 3.3. Bacterial Cultivation and In Vitro Antibacterial Assay

A single colony of *Escherichia coli* ATCC 25922 was inoculated into Luria Bertani (LB) liquid medium (HiMedia) (5 mL) at 35 °C for six hours. In a microtube, the bacterial inoculum was adjusted to a final concentration of 10^6^ CFU/mL; then, 50 mg of the glass or glass-ceramic sample sterilized (autoclaved) at 110 °C was transferred to the microtube. The antimicrobial activity was evaluated using two techniques, where the bacterial solution remained in contact with the sample for 0, 10, 15, 30, and 60 min for the agar incorporation test and 30 min for the optical density test. For the agar incorporation test, after the period of contact between sample and the inoculum, all the glass powder was transferred to a sterile Petri dish and homogenized together with a culture medium (LB agar, 40 °C) through circular movements. After solidification, the plates were incubated in the inverted position for 24 h at 35 °C. For the evaluation of the optical density in a liquid medium, after the period of contact of the sample with inoculum, 1 mL of liquid culture medium (LB) was added to the microtube, followed by homogenization. The suspension was centrifuged, and a 200 μL aliquot was transferred to a 96-well microplate. The samples were simultaneously incubated in the Microplate Reader LB940 (Berthold120 Technology, Bad Wildbad, Germany) under agitation at 35 °C for 24 h. Optical density was monitored at a wavelength of 600 nm. All experiments were performed in triplicate. The coding of the sample was defined as A = amber glass, P = flint glass, G = coarse particles, F = fine particles, T1 = thermal treatment at 700 °C, T2 = thermal treatment at 720 °C, and C = control experiment. Thus, the samples were named AGT1, AGT2, PGT1, AFT2, and PFT2.

## 4. Conclusions

In summary, we report an affordable approach to promoting antimicrobial activity in waste glass. Powdered glass waste can be converted into antimicrobial material with a thermal treatment under an air atmosphere without special requirements. It is worth mentioning that antimicrobial activity can be obtained in ordinary flint and amber glass waste. Furthermore, the time required to notice antimicrobial activity in glass-ceramics from waste flint glass is as short as 30 min for a material with high thermal and chemical resistance. The thermal treatment does not contribute to the segregation of any element or the severe loss of it. The antimicrobial activity was noticed after two hours of thermal treatment at 720 °C, and so, the nucleation of nanostructures was attributed to combeite and/or nepheline phases. After five hours of thermal treatment, the coexistence of combeite and nepheline phases was noticed in the glass-ceramic derived from glass waste. We hope these results will help support new studies on developing functionalities in glass waste.

## Figures and Tables

**Figure 1 molecules-29-03212-f001:**
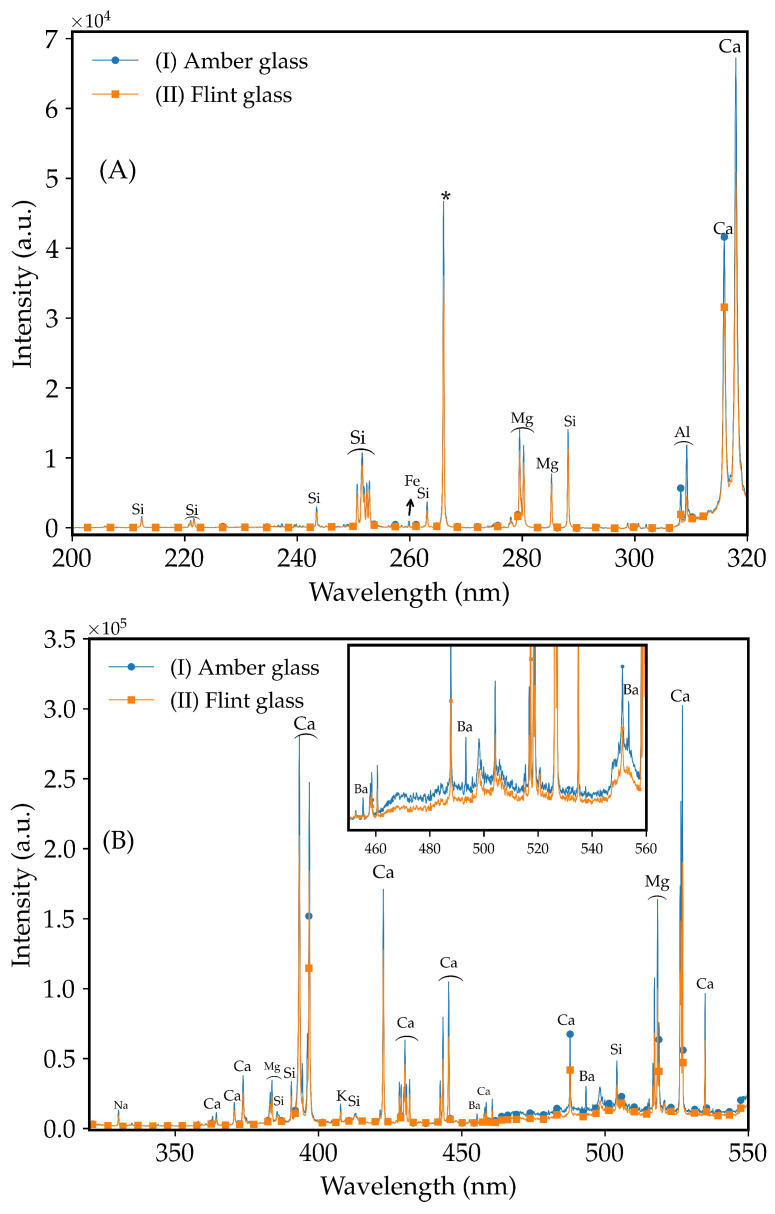
LIBS spectra for sintered (I) amber and (II) flint waste glass. Spectral range: (**A**) 200–320 nm ((*) denotes the laser line), (**B**) 320–555 nm (inset graphic: selected region showing Ba lines), and (**C**) 555–850 nm. Peaks under ⌢ (arc) symbol are attributed to the element above.

**Figure 2 molecules-29-03212-f002:**
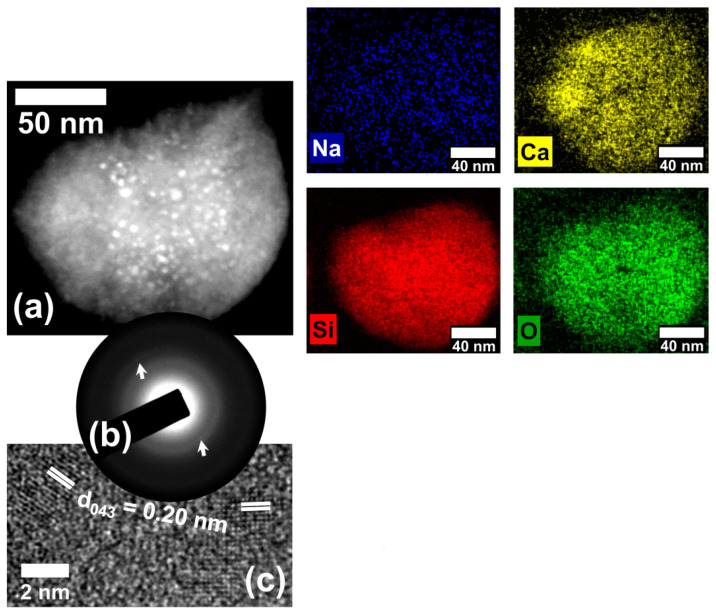
(**a**) ADF-STEM micrograph and a set of EDS elemental mappings of the sample annealed at 720 °C for 2 h. (**b**) is a SAED pattern from (**a**). (**c**) is an HRTEM image showing the (043) family planes attributed to the combeite phase.

**Figure 3 molecules-29-03212-f003:**
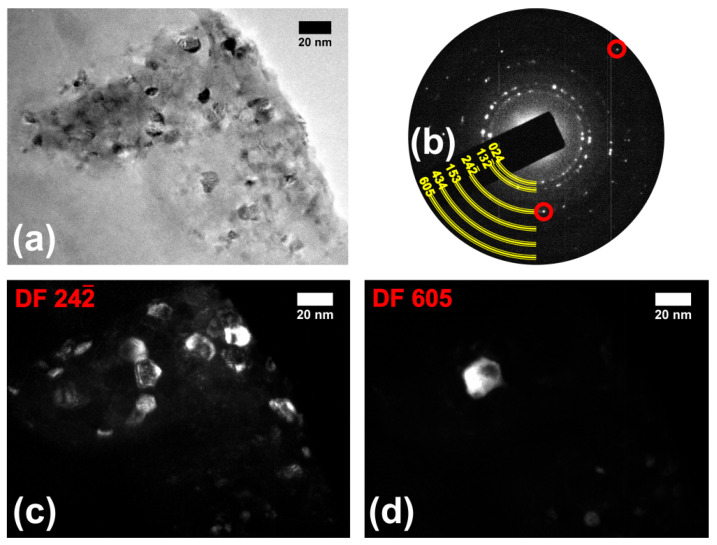
In (**a**), we have a classical TEM micrograph from the sample annealed at 720 °C for 5 h and (**b**) its SAED pattern; semirings represent the result of calculations based on the possible distances between crystalline planes indexed to combeite [(024), (13$\bar{2}$), and (24$\bar{4}$)] and nepheline [(103), (130), and (330)] phases. (**c**,**d**) are DF-TEM images obtained using the 24 and 605 diffracted beams selected in (**b**) [indicated by red solid circles], respectively.

**Figure 4 molecules-29-03212-f004:**
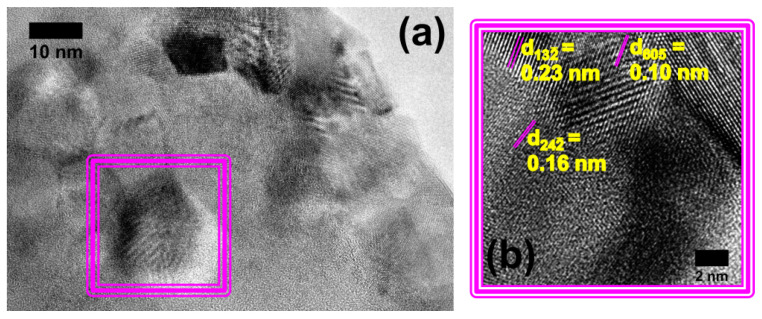
(**a**) High-resolution TEM micrograph from the sample annealed at 720 °C for 5 h and (**b**) higher magnification, revealing interplanar spacing of the coexistence of combeite and nepheline phases.

**Figure 5 molecules-29-03212-f005:**
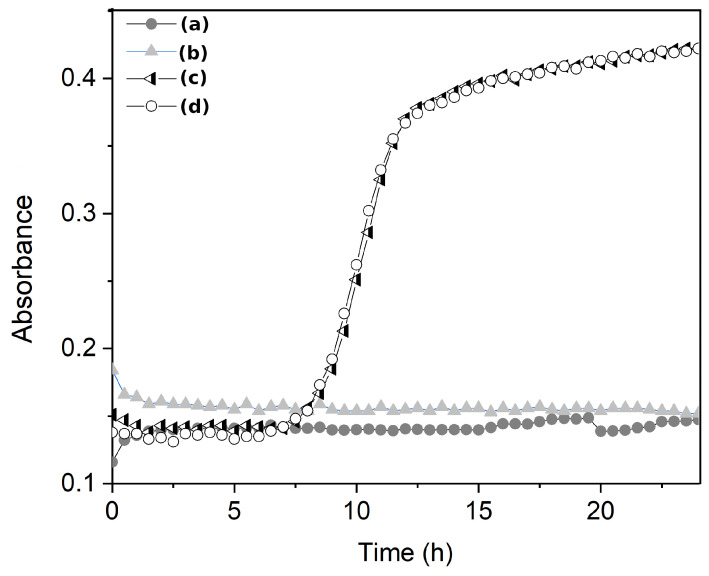
Optical density measurements of bacteria growth (*E. coli*, $10^{6}$ CFU) with a contact time of 30 min for thermally treated flint glass (720 °C during 2 h). Samples (**a**–**c**): flint glass (without thermal treatment); (**d**): control experiment.

**Figure 6 molecules-29-03212-f006:**
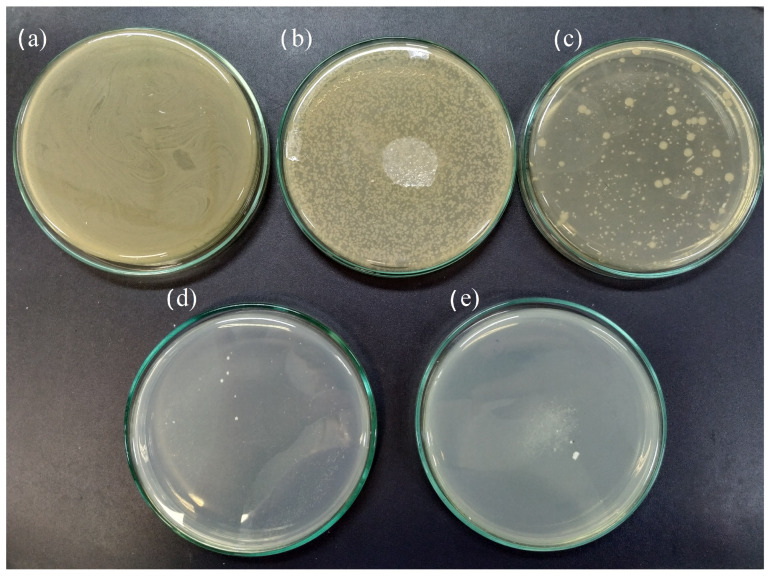
Bacterial growth of *E. coli* (10^6^ CFU) after contact time with the glass-ceramic of (**a**) 0 min, (**b**) 10 min, (**c**) 15 min, (**d**) 30 min, and (**e**) 60 min.

**Figure 7 molecules-29-03212-f007:**
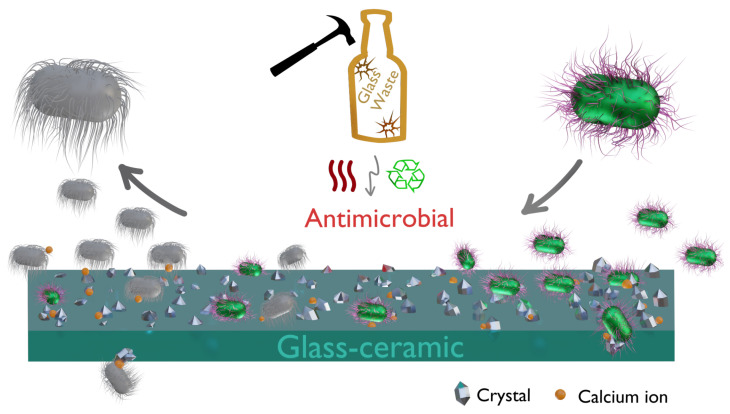
Schematic representation of the approach to obtaining glass-ceramics from glass waste and the proposed mechanism for antimicrobial activity.

**Table 1 molecules-29-03212-t001:** Antimicrobial activity assay and thermal treatment temperature.

Entry	${\bar{M}}_{s}$ ^(a)^	Treatment ^(b)^	Sample	CFU ^(c)^
	μm	°C		mL
Amber ^(d)^		-	CA	>3.0 $\times \;10^{4}$
Flint ^(e)^		-	CF	>3.0 $\times \;10^{4}$
Amber	322.6	700 °C	AGT1	6.1 $\times \;10^{3}$
		720 °C	AGT2	5.0 $\times 10^{3}$
Flint	326.5	700 °C	PGT1	8.8 $\times \;10^{3}$
		720 °C	PGT2	0
Amber	179.9	700 °C	AFT1	1.1 $\times \;10^{4}$
		720 °C	AFT2	7.6 $\times 10^{3}$
Flint	199.5	700 °C	PFT1	1.9 $\times \;10^{4}$
		720 °C	PFT2	8.4 $\times \;10^{3}$

^(a)^ Mean size (${\bar{M}}_{s}$); ^(b)^ thermal treatment during 2 h; ^(c)^ colony-forming unit; ^(d)^ control experiment amber glass; ^(e)^ control experiment flint glass.

## Data Availability

Data are contained within the article.

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
