# Peer review of "Nanostructured Glass-Ceramic Materials from Glass Waste with Antimicrobial Activity"

_molecules, 2024, doi:10.3390/molecules29133212_

Round 1

Reviewer 1 Report

Comments and Suggestions for Authors

"...the presence of the combeite phase together with the glass matrix facilitates the release of calcium ions (Figure 4), which in turn leads to membrane depolarization and the subsequent death of the cells [10]".

If the cause of the antibacterial effect is calcium, then the whole experiment is pointless.

Author Response

We appreciate the referee's insightful observation regarding the potential role of calcium in the observed antimicrobial activity.  We acknowledge that calcium and its compounds are known to exhibit antimicrobial effects in certain contexts. However, we respectfully disagree with the assertion that this renders the entire experiment pointless.

Our study presents several key findings that go beyond simply demonstrating antimicrobial activity:

  1.   Novel Material Synthesis: The primary focus of our research is to introduce a novel and sustainable method for transforming waste glass into functional materials. The antimicrobial properties, while significant, are a secondary outcome of this primary goal.
  2.   Mechanistic Exploration: We conducted a thorough analysis of the chemical and structural transformations occurring during the thermal treatment. This analysis reveals the formation of specific crystalline phases (combeite and nepheline) that may contribute to the antimicrobial activity, in addition to any potential calcium-related effects.
  3.   Comparative Analysis: We investigated both flint and amber-colored glass waste, which exhibit different chemical compositions (including varying calcium content). The observation of antimicrobial activity in both materials suggests that the effect is not solely attributable to calcium.

Additionally, as reported by Cabal et al (ref 10 of the manuscript, DOI:10.1038/srep05440) the biocidal activity arises from calcium release at the glass-particle interface. Thus, glass-membrane interface is crucial for the biocide activity being the local high calcium ions concentration responsible for cell death (DOI: 10.1002/adem.201080133;  line 231-232 of new manuscript version or line 172-173 of the original submission). A similar conclusion is obtained in dental materials (DOI: 10.3390/antibiotics10070865)(DOI: 10.1038/s41598-019-55288-3).

Reviewer 2 Report

Comments and Suggestions for Authors

Dear Authors,

In this paper, the authors showed an affordable route to convert powdered waste glasses into glass-ceramic materials with antimicrobial activity. The results indicated that powdered glass waste can be converted into antimicrobial material after two hours of thermal treatment at 720 ℃. Furthermore, the time required to notice antimicrobial activity in glass ceramics from waste flint glasses is as short as 30 min for a material with high thermal and chemical resistance. The manuscript is very interesting, but some points should be addressed before the manuscript is considered for publication. So, the manuscript is accepted after minor revision.

In the Introduction section:

Please re-write the introduction by adding more references related to the work of antimicrobials from glass-ceramic materials.

In the Methods section:

It is better to rearrange the experiment as follows:

1- Antimicrobial glass-ceramic synthesis.

2- Morphological and spectroscopy characterizations.

3- Bacterial Cultivation and In Vitro Antibacterial Assay.

In the Results and Discussion section:

Please rearrange the results and discussion as the above suggestion in the methods section and transfer Table (1) from the method section to the result and discussion section.

The above minor remarks do not reduce the quality of the Manuscript, and after addressed, it can be published in the journal.

Thank you very much

Author Response

The changes were performed as suggested.

We have now added the following lines in the introduction Section (lines 69-72). 

Furthermore, in dentistry, root canal filling is performed with tricalcium silicate-based materials that promote not only antimicrobial activity but also act as a bioactive material that can induce the formation of hard tissue. As a general trend the calcium release is positively correlated with enhanced antimicrobial performance.[13,14]

Two references were added 

  1. Janini, A.C.P.; Bombarda, G.F.; Pelepenko, L.E.; Marciano, M.A. Antimicrobial Activity of Calcium Silicate-Based Dental Materials: A Literature Review. Antibiotics 2021, 10, 865. https://doi.org/10.3390/antibiotics10070865.
  2. 14. Koutroulis, A.; Kuehne, S.A.; Cooper, P.R.; Camilleri, J. The role of calcium ion release on biocompatibility and antimicrobial properties of hydraulic cements. Scientific Reports 2019, 9. https://doi.org/10.1038/s41598-019-55288-3.

Reviewer 3 Report

Comments and Suggestions for Authors

The manuscript contains interesting results regarding an approach to enhance the antimicrobial activity of waste glass by converting it into an antimicrobial material through heat treatment in an air atmosphere. The manuscript can be published after correction

Remarks

1. It should be indicated what the arcs mean in Figures 1 (A - C)

2.  The accuracy of the values ​​in Figures S1 - S4 should be indicated. What is the error of the experiment for these figures

3. Line 199: "The powder X-ray diffraction analyses (Figure S5, Supplementary Information) of annealed (Figure S5 (i)) and unannealed (Figure S5 (ii)) glass powder waste do not show any peak which can be associated with the glass crystallization, probably due to the relatively low amount of the crystalline phases" -

The XRD patern shown in Fig. S5 looks weird. In my opinion, this is a typical x-ray scattering curve from a heterogeneous structure. In order to dmake conclusion regarding absence of crystallization of the amorphous phase, the authors must indicate the half-widths of the diffuse peak and discuss its value. The diffuse peak is very similar to the superposition of two peaks from two amorphous phases with similar radii of the first coordination sphere

Author Response

  1. The “arcs” indicates that peaks under this symbol (arc) are attributed to the element above the arc.

We have now added the following line in the caption (Figure 1). 

"...Peaks under ဂ (arc) symbol are attributed to the element above."

  1. The measurements of the particle size distribution (PSD) (Figures S1-S4) were performed in duplicate. The results of the geometric standard deviation are: Figure S 1(coarse amber glass) = 1.5 μm, Figure S 2(coarse flint glass) = 1.4 μm,  Figure S 3(fine amber glass) = 1.6 μm, and  Figure S 4(fine flint glass) = 1.5 μm. These values were added in the caption of the figures in supplementary information. 
  2. We appreciate the referee's observation regarding the XRD pattern in Figure S5. We agree that the pattern exhibits the typical broad, diffuse halo characteristic of amorphous materials. As the referee correctly notes, the presence of crystalline phases within an amorphous matrix would typically manifest as sharper peaks superimposed on this diffuse background [https://doi.org/10.1021/cr60084a007].

However, we respectfully disagree with the characterization of the pattern as solely representing X-ray scattering from a heterogeneous structure. The width of the diffuse halo (approximately 15° 2θ) is consistent with amorphous materials and does not necessarily imply the presence of distinct crystalline phases. While peak broadening due to particle size effects can occur, as described by the Scherrer equation, this typically results in a broadening of a few degrees, not the extensive broadening observed here. Furthermore, to address the referee's comment more directly, we have now included an analysis of the full width at half maximum (FWHM) of the diffuse peak in the supplementary information (in the figure caption of Figure S5 ). The FWHM values are within the range expected for amorphous materials, further supporting our conclusion regarding the absence of significant crystallization in the glass powder waste samples.

We have rewrite the following sentence from (line 168-169)

The powder X-ray diffraction analyses (Figure S5, Supplementary Information) of annealed (Figure S5 (i)) and unannealed (Figure S5 (ii)) glass powder waste do not show any peak which can be associated with the glass crystallization, probably due to the relatively low amount of the crystalline phases.

To 

Powder X-ray diffraction analyses of both annealed (Figure S5(i)) and unannealed (Figure S5(ii)) glass powder waste did not reveal any peaks indicative of glass crystallization, probably due to the low abundance of crystalline phases.

Reviewer 4 Report

Comments and Suggestions for Authors

The topic of the manuscript presented “Nanostructured glass-ceramic materials from glass waste with antimicrobial activityin this work is a distinctly up-to-date topic and a good approach aimed to help moving forward in efforts to tackle the global problems of pollution and waste recycling. Тhe added value of the research comes from the antimicrobial properties of the new material that the authors of the paper managed to achieve during their research. Materials and methods used are well described, which will help other researchers to repeat their experiments and compare theirs.

After reading the manuscript, a few questions arise:

1. The authors describe their method for treatment of waste glass as affordable. Conventional glass recycling methods are presented as logistics challenging and with height cost. In the manuscript there is no evaluation and explanation how and why the method they present can be more easily and cheaply implemented at remote areas and solve this problem. Please add some technical and cost comparison.

2. The sentence at line 52 “In this sense, the antimicrobial character of glasses is unusual. The synthesis of antimicrobial glasses/glass-ceramic from ordinary silica-based glasses can take the chemical resistance and an affordable source of the material.” is unclear.

3. It would be beneficial to add information about the physical properties of the recycled glass thus obtained. This would help in assessing its applicability.

4. What is the endurance over time of the antibacterial properties of the material obtained?

5. How and why a 2-hour heat treatment duration was chosen. With a shorter processing time, are the antibacterial properties present?

6. Antimicrobial activity assay for thermal treatment of 5 hours was not discussed in the text.

A better and more detailed explanation of the treatment parameters chosen in the experiment is needed.

The figures are located in relation to the text too far from the paragraph  they are commented. This makes it difficult to read and follow the results.

Comments on the Quality of English Language

English language used in the article is easy to `read and understand.

Author Response

Our answers:

  1. We thank the referee for the observation. As mentioned in the introduction section (lines 44-47), not only the development of strategies for the reuse of waste glasses as new materials are desirable, but also adding functional properties to the waste-based material is important. The production of new materials, for example new bottles, only from glass waste requires several approaches from logistics to quality/amount of the glass waste type. For example, green and amber glasses have a relatively wide mixture of colors (respectively up to 95% and 70%), while only 60% is acceptable for flint/white glass production from waste. Thus, not only the color of the glass waste can be a problem for recycling, but also the appropriate waste amount of each glass type that can be used for glass production from waste (http://dx.doi.org/10.1016/b978-0-12-396459-5.00014-3). As mentioned, logistics, differences in government policy, consumer education, and habits affect recycling. In other words, glass recycling requires not only technical viability, but changes in the political society's efforts.  The recycling of glass can be performed in a closed-loop, producing new waste bottles, for example, or in an open-loop where the glass is, usually, remelted but involve formation of different products as bricks, foams, aggregate, abrasives, porcelain, and glass-ceramics. In this sense, the approach proposed, an open-loop, does not require any additional reagents or addition of materials to obtain the new glass-based product, just a thermal annealing of the glass waste with an ordinary furnace operating in relatively mild conditions. 
  2. We thank the referee for the observation. Silica-based glasses are recognized by thermal and chemical resistance. The thermal treatment proposed in this work provides just crystallization of nanocrystals. Thus, it is expected that, due to the high temperature synthesis of the glass, the glass-ceramics obtained after thermal treatment can show the chemical resistance of the pristine material. Moreover, glasses show high thermal and chemical resistance with low cost, which can be obtained at large scale worldwide. The relatively low cost of the glass materials, principally after their use, results in a low income for recycling. Thus, the availability of the glasses in waste becomes an opportunity for the development of new materials or new functionalities.  

The changes in the introduction Section are detached in blue. We now rewrite the sentence:

In this sense, the antimicrobial character of glasses is unusual. The synthesis of antimicrobial glasses/glass-ceramic from ordinary silica-based glasses can show the chemical resistance of the pristine material. Moreover, glasses show high thermal and chemical resistance with low cost which can be obtained at large scale worldwide. The relatively low cost of the glass materials, principally after their use, results in a low income for recycling. Thus, the availability of the glasses in waste becomes an opportunity for the development of new materials/functionalities in an open-loop recycling.

3. We thank the referee for encouraging us to enlarge the uses of the material. The antibacterial properties  of the glass-ceramics were evaluated after 6 months, with the same activity. Some new practical properties/approaches are under evaluation, such as the possibility of thermal deposition of the glass waste with further thermal treatment. Moreover, it is under evaluation the development of antimicrobial activity of glass samples with micro roughness.

4. We thank the referee for the observation. Silica-based glasses are recognized by thermal and chemical resistance. The thermal treatment proposed in this work provides just crystallization of nanocrystals. Thus, it is expected that  due to the high temperature of the glasses  the glass-ceramics obtained after thermal treatment can show the chemical resistance of the pristine material. Moreover, glasses show high thermal and chemical resistance with low cost which can be obtained at large scale worldwide. The relatively low cost of the glass materials, principally after their use, results in a low income for recycling. Thus, the availability of the glasses in waste becomes an opportunity for the development of new materials or new functionalities.  

5. In the screening evaluation, we noticed that it is required 2 hours of thermal treatment to obtain the biocidal effect. The crystallization process requires not only a minimum temperature (or a temperature where reaction rate is maximum, expressed by Kissinger equation (10.1021/ac60131a045)) of the thermal treatment, but also is necessary to hold the glass at specific temperature to obtain the glass crystallization (10.1016/j.matchemphys.2005.09.060). The thermal treatment was selected performing a screening of the temperatures used for crystallization of silica-based glasses. 

We now added the following sentence in the experimental section (subsection 2.1):

The thermal treatment was selected performing a screening of the temperatures usually employed for crystallization of silica-based glasses [10,17]. 

6. The thermal treatment of 5 hours was employed to obtain a more crystallized sample and improve the SAED results. As the antimicrobial activity was noticed with just 2 hours requiring less time/energy consumption the antimicrobial assays were performed with this set of samples.